# Genome-Wide Identification and Expression Analysis of *Bx* Involved in Benzoxazinoids Biosynthesis Revealed the Roles of DIMBOA during Early Somatic Embryogenesis in *Dimocarpus longan* Lour

**DOI:** 10.3390/plants13101373

**Published:** 2024-05-15

**Authors:** Xiaoqiong Xu, Chunyu Zhang, Chunwang Lai, Zhilin Zhang, Jiajia Wu, Qun Su, Yu Gan, Zihao Zhang, Yukun Chen, Rongfang Guo, Yuling Lin, Zhongxiong Lai

**Affiliations:** Institute of Horticultural Biotechnology, Fujian Agriculture and Forestry University, Fuzhou 350002, China; xuxq0921@163.com (X.X.); zcynhba@163.com (C.Z.); laichunwang@163.com (C.L.); 17711828555@163.com (Z.Z.); wjj070329@163.com (J.W.); qunsu315@yeah.net (Q.S.); ganyu0514@163.com (Y.G.); zhangzihao863@126.com (Z.Z.); cyk68@163.com (Y.C.); guorofa@163.com (R.G.); buliang84@163.com (Y.L.)

**Keywords:** *Dimocarpus longan* Lour., benzoxazinoids, *Bx* gene family, DIMBOA, IAA, somatic embryogenesis

## Abstract

Benzoxazinoids (BXs) are tryptophan-derived indole metabolites and play a role in various physiological processes, such as auxin metabolism. Auxin is essential in the process of somatic embryogenesis (SE) in plants. In this study, we used bioinformatics, transcriptome data, exogenous treatment experiments, and qPCR analysis to study the evolutionary pattern of *Bx* genes in green plants, the regulatory mechanism of *DlBx* genes during early SE, and the effect of 2,4-dihydroxy-7-methoxy-1,4-benzoxazine-3-one (DIMBOA) on the early SE in *Dimocarpus longan* Lour. The results showed that 27 putative *DlBxs* were identified in the longan genome; the *Bx* genes evolved independently in monocots and dicots, and the main way of gene duplication for the *DlBx* was tandem duplication (TD) and the *DlBx* were strongly constrained by purification selection during evolution. The transcriptome data indicated varying expression levels of *DlBx* during longan early SE, and most *DlBxs* responded to light, temperature, drought stress, and 2,4-dichlorophenoxyacetic acid (2,4-D) treatment; qRT-PCR results showed *DlBx1*, *DlBx6g* and *DlBx6h* were responsive to auxin, and treatment with 0.1mg/L DIMBOA for 9 days significantly upregulated the expression levels of *DlBx1*, *DlBx3g*, *DlBx6c*, *DlBx6f*, *DlB6h*, *DlBx7d*, *DlBx8*, and *DlBx9b*. The correlation analysis showed a significantly negative correlation between the expression level of *DlBx1* and the endogenous IAA contents; DIMBOA significantly promoted the early SE and significantly changed the endogenous IAA content, and the IAA content increased significantly at the 9th day and decreased significantly at the 13th day. Therefore, the results suggested that DIMBOA indirectly promote the early SE by changing the endogenous IAA content via affecting the expression level of *DlBx1* and hydrogen peroxide (H_2_O_2_) content in longan.

## 1. Introduction

Benzoxazinoids (BXs) are tryptophan-derived indole metabolites commonly found in cereals such as maize (*Zea mays*), wheat (*Triticum aestivum* spp.) and rye (*Secale cereale*) [1] and are less prevalent in dicots [2]. They can be categorized into hydroxamic acids, lactams, benzoxazolinones and methyl derivatives. Among these, DIMBOA is the most widely studied BXs substance. BXs have insecticidal, disease resistance, and allelopathic effects, and also play a role in regulating the flowering time, auxin metabolism, and rhizosphere microbial community [3]. Studies have shown that BXs can defend against pests such as aphids, *Spodoptera litura* and *Pyraustanubilalis*, either through direct toxicity or anorectic effects [4,5]. BXs also have toxic effects on plants, inhibiting the germination, growth and development of weeds such as *Amaranthus retroflexus* and *Portulaca oleracea* L. [6,7]. Research has shown that maize plant root exudates containing BXs can drive soil responses to plant growth and defense by shaping the rhizosphere microbiota [8]. Genetic analysis has revealed that *Bx12* affects corn flowering time [9]. BXs have been recognized as inhibitors of auxin that regulate phototropism in maize, but it has also been proposed that the primary factor affecting the growth and decreased curvature of maize embryonic sheaths was due to the release of H_2_O_2_ from DIMBOA produced on the light-exposed side [10]. As a result, the relationship between BXs and auxin remains highly disputed. Therefore, it is practically significant to gain a deeper understanding of the biosynthetic process of plant BXs and to investigate more genes related to plant BXs biosynthesis and their evolutionary process.

Dicotyledonous plants that produce benzoxazinoids are not considered as model plants, and there are no genetically defined lines available. Previous research [2] has shown that CYP71C enzymes are involved in the biosynthesis of DIBOA (2,4-dihydroxy-1,4-benzoxazin-3-one) in *Consolida orientalis* and *Lamium galeobdolon*, and that an UDP-glucosyltransferase (UGT) generates the less toxic benzoxazinoid-glucoside, similar to what occurs in grasses. Hence, in principle, the benzoxazine biosynthesis pathway seems to follow the same chemical route in both dicots and monocots. At present, the biosynthesis pathways, key enzymes and related gene functions of BXs in maize have been extensively studied [11,12], and the related genes involved in the BXs biosynthesis pathway were named *Bx* genes. Firstly, *TSA* and *Bx1* are *indole-glycerolphosphate lyases* (*IGL*). *Bx1* converts indole-3-glycerolphosphate (IGP) to indole in chloroplasts. Then, in the endoplasmic reticulum, indole is successively oxidized to DIBOA by four members of the *CYP71C* subfamily (*Bx2–5*) of cytochrome P450 mono-oxygenases [13]. Subsequently, in the cytoplasm, two *UGTs Bx8* and *Bx9* glycosylated DIBOA to generate stable DIBOA-Glc to avoid the autotoxicity of BXs [14]. DIBOA-Glc is successively oxidized and methylated in the cytoplasm by *Bx6* (*2-oxoglutarate-dependent dioxygenase*, *2-ODD*) and *Bx7* (*O-methyltransferase*, *OMT*) to generate DIMBOA-Glc [15]. DIMBOA-Glc is converted to HDMBOA-Glc (2-hydroxy-4,7-dimethoxy-1,4-benzoxazin-3-one glucoside) by the action of 4 *OMTs* (*Bx10–12*, *Bx14*). In addition, *Bx13* (*2-ODD*) catalyzes the synthesis of TRIMBOA-Glc (2,4,7-trihydroxy-8-methoxy-1,4-benzoxazin-3-one glucoside) from DIMBOA-Glc, which is then catalytically converted to HDIM_2_BOA-Glc (2-dihydroxy-4,7,8-trimethoxy-1,4-benzoxazin-3-one glucoside) by the successive action of *Bx7* and *Bx14* [16]. BXs are generally stably stored as glycosides in vesicles to avoid autotoxicity for the plant itself. Their defensive function depends on the temporal and spatial activation of specific hydrolases. When the plant is subjected to stress or cellular disruption, BXs-Glc is hydrolyzed on contact with β-glucosidase, which eliminates the glycoside and releases toxic BXs, thus acting as an autotoxic defense [11,16].

Longan, belonging to the Longan genus in the Sapindaceae family, is an evergreen plant native to temperate Asia and the tropics, and is extensively cultivated in southern China and Southeast Asia [17]. The development of longan embryos has a significant impact on fruit production and quality, and in-depth research into the mechanism of longan embryo development is crucial for the longan industry [18]. However, studying longan embryogenesis in its natural state has many drawbacks, such as sampling difficulties, poor material synchronization and uncontrollable development process, etc. The longan somatic embryogenesis (SE) system established by Lai and others [19,20], which has the characteristics of high synchronization, frequency, and regeneration capacity, can solve the above-mentioned problems very well. It is well known that most plant somatic embryos are induced by auxin [21], and SE is regulated by adding different concentrations of auxin [22,23]. During longan SE, relatively high levels of endogenous IAA underpin the induction and maintenance of embryonic cytogenesis, and changes in endogenous IAA levels affect the development of longan somatic embryos [24]. The available evidence suggests that there are two pathways for de novo auxin biosynthesis (tryptophan-dependent and tryptophan-independent) in plants [25], indole-3-glycerolphosphate (IGP) is a branch of these pathways [26] and an important precursor of the BXs biosynthesis pathway. Previous studies have shown that ZmAuxRP1 may function as a key regulator of resource redistribution between IAA and DIMBOA, contributing to the balanced growth and disease resistance of maize by controlling IGP flow [27]. Our lab’s earlier work has revealed the presence of BXs compounds, specifically DIMBOA-Glc and TRIBOA-Glc, in longan embryonic cultures. In addition, Hansheng Li [28] studied longan EC with different light qualities (dark, blue, and white light), and performed KEGG enrichment analysis on the target genes of differentially expressed miRNAs, and the results showed that the BXs biosynthesis pathway was significantly enriched under blue light and white light treatment. According to a large number of studies on BXs compounds in grass plants, DIMBOA is the main resistance and allelopathic substance in maize, which has the effect of affecting plant root growth and participating in auxin metabolism. In conclusion, it is suggested that DIMBOA may affect the early stages of longan SE and has some relationship with auxin. However, the effects of DIMBOA on the SE process and its relationship with IAA remain unclear in longan. In addition, studies on genes related to the BXs biosynthesis pathway of longan (*DlBx*) have not been reported. In this study, a total 27 *DlBx* genes were identified and the basic physical and chemical properties, phylogeny, gene structure, motif compositions, chromosome distribution, cis-acting elements and transcription factors were analyzed. Combined with transcriptome data, the expression patterns of *DlBx* in the early stages of longan SE (embryogenic callus (EC), incomplete embryotic compact structure (ICpEC) and globular embryo (GE)) were analyzed under different treatments. At the same time, we investigated the effects of DIMBOA on the morphogenesis and endogenous IAA content changes in early longan SE, as well as the expression of key *DlBx* genes. This was performed in order to provide a theoretical reference for the genomic information and gene function verification and utilization of *DlBx* gene family members.

## 2. Results

### 2.1. Identification of Benzoxazinoids Biosynthesis (Bx) Genes in Plant

The key genes involved in the benzoxazinoid biosynthesis pathway of *Z. mays* (*ZmBx1–ZmBx14*) were used as bait to identify *Bx* homologous genes in 24 species. These species were categorized into six groups: Algae (2), Bryophyta (2), Pteridophyta (2), Gymnospermae (2), Monocots (6) and Dicots (10). Based on the research on the biosynthesis pathway of BXs in maize, it was found that 14 *Bx* genes belonged to 5 gene families. Specifically, *Bx1* was identified as a homolog of *TSA*, *Bx2–5* were part of the *CYP71C* subfamily, *Bx8* and *Bx9* belonged to the *UGT* family, *Bx6* and *Bx13* belonged to the *2-ODD* family, and *Bx10–Bx12* and *Bx14* were part of the *OMT* family.

A total of 363 non-redundant *Bx* genes were obtained from 24 species, with the number of homologous genes of *Bx* genes in each species ranging from 1 to 29 (Appendix A). Among them, there were 27 candidate genes in longan, and *Bx1*, *Bx2–5*, *Bx6/13*, *Bx7/10–12/14*, and *Bx8/9* have 1, 10, 8, 5, and 3 members, respectively. Based on their identity with the *ZmBx* gene and their chromosomal position, they were named *DlBx1*, *DlBx2*, *DlBx3a–g*, *DlBx4a–b*, *DlBx6a–h*, *DlBx7a–e*, *DlBx8*, *DlBx9a–b*. *Bx1* and *IGL* divert IGP and/or indole from primary metabolism (Trp) toward secondary metabolism (BXs) [11]. In this study, only one *IGL* gene (*DlBx1*) was identified in longan, unlike the three *IGL* genes (*ZmBx1*, *ZmIGL1*, and *ZmTSA1*) that were present in maize. Therefore, *DlBx1* plays an important role in the trp and BXs biosynthesis pathways. In each group, the total number of *Bx* homologous genes was arranged in descending order as follows: dicots > monocots > gymnosperms > ferns, and only one homologous gene existed in algae and bryophytes. The number of members of each gene family varied greatly among the different species (Figure 1A). For example, in monocots, six, six, and three, *Bx1* were identified in *Brachypodium distachyon*, *Oryza sativa*, and *Triticum aestivum*, respectively, while only one *Bx1* was retained in, *Hordeum lechleri*, *Secale cereale*, *Zea mays*. Among the ferns, three and two members were identified in *Ceratopteris richardii* and *Selaginella moellendorffii*, respectively. In gymnosperms, *Bx1* was only present in *Picea abies* and two members were identified. The copies of the *Bx1* gene were identified in dicots ranging from 1 to 3. Compare the average number of genes in each group (Figure 1B). The results showed that the average number of *Bx1* homologous genes followed an ‘M’ shape during plant evolution, reaching a peak in ferns and monocots, indicating that *Bx1* homologous genes gradually expanded or lost during evolution. However, the average number of *Bx2–5*, *Bx6/13*, *Bx7/10–12/14* and *Bx8/9* homologous genes gradually increased with plants evolution, suggesting that there may have been an expansion in the process of plant evolution. The physicochemical properties of the longan DlBx protein, including amino acid number (aa), molecular weight (MW), theoretical isoelectric point (pI), instability index (II), the grand average of hydropathicity (GRAVY), TMHs and subcellular localization, were analyzed, and the results are shown in Appendix A.

### 2.2. Phylogenetic Tree and Genome Structure Analysis of Benzoxazinoids Biosynthesis Genes in Green Plants

Phylogenetic analyses were conducted for each gene family to investigate the evolutionary relationships among genes related to benzoxazinoid biosynthesis (Figure 2, Appendix A). In the *Bx1* gene family (Figure 2), the established tree consisted of two major branches, in addition to two algae homologs (Crei_XP_001696973.2 and Otau_XP_003078551.1) and two mosses homologs (Mpol_PTQ31337.1 and Ppat_Phpat.004G068200) that were positioned as an out-group to the evolutionary origin of Bx1 proteins in plants. Bx1 proteins can be divided into three types (I–III) based on their sequence similarity in plants. Among them, fern Bx1 proteins (I) and monocot Bx1 proteins (II) were clustered separately, while dicot Bx1 proteins and *P. abies* were clustered together (III). Notably, the Pabi_MA_616184g0010 is located in the outer layer of the monocot and dicot clade, suggesting that this Bx1 protein may represent the ancient out-group of the higher plant Bx1. Bx1 proteins were found in all tested green plants (except *P. taeda*) in this study, indicating that the divergence of Bx1 protein in green plants may have occurred in the common ancestral species of extant green plants, suggesting that these proteins may have an essential physiological role for these plants. The phylogenetic tree of Bx2–5 homologs (Appendix A) was also composed of two main branches, except for the Bx2–5 protein of the fern *C. richardii*, which were located at the evolutionary origin. The phylogenetic tree of Bx2–5 homologs were also clearly divided into three types: the Bx2–5 homologs of monocots were clustered separately (I), the Bx2–5 homologs of dicots could be divided into two types, and a small number of Bx2–5 homologs of dicots and gymnosperms were clustered into one class (III), and the rest were clustered separately (II). Notably, the longan DlBx4a protein existed before the divergence of monocots and dicots, and it was speculated that the evolutionary origin of DlBx4a may be earlier. In the phylogenetic tree of Bx7/10–12/14 proteins (Appendix A), Ptri_XP_002305103.1, Ptri_XP_006377432.2, and Mdom_XP_028959002.1 were located in the outer layer, and the remaining Bx7/10–12/14 proteins could be divided into two types, in which the Bx7/10–12/14 homolog of dicots were clustered separately (I), while the Bx7/10–12/14 homologs of monocots and gymnosperms were clustered together in one class (II). Bx6/13 homologs and Bx8/9 homologs were only present in angiosperms, and both Bx6/13 and Bx8/9 homologs in monocots were located in the outer layer of dicots, speculating that Bx6/13 and Bx8/9 homologs might have evolved vertically within angiosperms, and their divergence occurred in the common ancestor of angiosperms. The phylogenetic tree of the Bx6/13 homolog (Appendix A) could be divided into three types, of which monocots were clustered separately (I) and the dicots could be divided into two groups (II and III). In longan, DlBx6a and DlBx6b were clustered into one class (III), and DlBx6c h were clustered into one class (II). The phylogenetic tree of the Bx8/9 homolog (Appendix A) could be divided into two types, in which monocots (I) and dicots (II) were clustered separately. Furthermore, the five gene families of the monocots and dicots could be clearly distinguished.

Previous studies have shown that genetic structural diversity plays an important role in driving the evolution of gene families [29]. In order to further understand the potential relationship between the gene structure and evolutionary history of the *Bx* gene family in plants, the gene structure of the *Bx* gene family was analyzed according to the classification of the established phylogenetic tree. Initially, 20 potential motifs were analyzed for proteins from five gene families using MEME v5.5.5 online software (Figure 2, Appendix A). The results showed that there were similar and specific conserved motifs in Bx1 proteins (Figure 2). For example, motif 10, motif 12–14, motif 16, and motif 18 were only found in type II Bx1, motif 8, motif 11, motif 15, and motif 19 are specific motifs in type III Bx1, and more than 80% of Bx1 proteins had motif 1–7. It was worth noting that the Bx1 protein Pabi MA 101664g0010 of *P. abies* has only motif 2, motif 5 and motif 9, and Pabi MA 616184g0010 has only motif 1 and motif 6. All Bx2–5 proteins have motif 1–4, 6, 8, 10–12, and 14–15. These motifs were similar in location and length (except Nl01g03270.mrna1) (Appendix A). It could be seen that the Bx2–5 protein was highly similar in structure, and it was structurally conserved in the evolutionary process. However, there were also specific motifs for the Bx2–5 protein in monocots and dicots, i.e., motif 16 was only present in monocots, and motif 18 and motif 20 were only found in dicots. Among the Bx6/13 proteins (Appendix A), motif 1–3, 5–7, 9 and 13 were conserved motifs of plant Bx6/13 proteins, and their distribution positions and lengths were similar, while motif 8 and 18–20 were only found in dicots. In general, the Bx6/13 protein was structurally conserved during evolution, but its functions also diverged after the differentiation of monocots and dicots. Among the Bx7/10–12/14 proteins (Appendix A), motif 1, 3, and 7–8 were present in the Bx7/10–12/14 proteins of all tested plants, and more than 85% of the Bx7/10–12/14 proteins had motif 2, 4–5, and 9–12. In addition, motif 14 was only found in the Bx7/10–12/14 protein of monocots, and motif 16–17 and motif 20 were only found in the Bx7/10–12/14 protein of dicots. Among the Bx8/9 proteins (Appendix A), motif 1–5, motif 8–9, and motif 11–13 were present in all tested plant Bx8/9 proteins, and more than 90% of Bx8/9 proteins had motif 6–7, motif 10, and motif 14–16. In addition, motif 9 was only found in the Bx8/9 protein of monocots, and motif 17–18 was only found in the Bx8/9 protein of dicots. These results suggest that the Bx protein sequence and function of different taxa in the five BXs biosynthesis-related gene families have changed greatly during evolution, and differentiation has occurred among the 24 tested species.

Intron/exon number data for five gene families were generated from the genome annotation files in order to gain a deeper understanding of the structural diversity within these gene families (Appendix A). Due to the lack of genomic annotation information for *H. Lechleri*, it is excluded from this analysis. In the *Bx1* gene, the number of exons ranges from 1 to 10. Among them, type I *Bx1* had a large number of exons, ranging from 7 to 10. In type II *Bx1*, six exons (5 *Bx1*) were the most common, followed by seven and eight exons (4 *Bx1*). In type III *Bx1*, nine exons (11 *Bx1*) were the most common, followed by eight exons (5 *Bx1*), in addition to the low number of exons in some *Bx1* genes in this type, suggesting that there may have been exon loss events during evolution. Among the 128 *Bx2–5* genes, 94 *Bx2–5* genes had two exons, which once again verified that the *Bx2–5* genes were structurally conserved. In the type I *Bx2–5* genes, most homologous genes contained three exons, followed by two exons. In addition, in the type I and type II *Bx2–5* genes, a small number of members contained four and ten exons. In the *Bx6/13* gene, the exon range was between one and five. Among them, the number of exons in the type I *Bx6/13* gene was small, and one exon was the most common. Type II and Type III genes have a relatively high number of exons, with three exons being the most common. The number of exons in the *Bx7/10–12/14* genes ranges from two to five, with two exons being the most common. In the *Bx8/9* gene, the number of exons ranges from two to six. Among them, the *Bx8/9* gene (type I) of monocots contained two exons except for *Osat XP 015617736.1*. The *Bx8/9* gene (type II) of dicots contains five and six exons in *Smuk GWHPBECP004864* and *Lchi LITCHI009263.m1*, respectively, suggesting that there may be exons during evolution. In conclusion, there are large variations in the number of exons or introns between different taxonomies of the phylogenetic tree of the five gene families or between the same taxonomics, and it is suggested that the function of the *Bx* gene may be differentiated through loss or gain events during evolution. Genes within the same gene family had similar intron/exon structures. Based on the presence or absence of introns, eukaryotic genes are classified as intronless, intron-poor (three–four introns), or intron-rich genes [30]. As can be seen, most *Bx2–5*, Bx6/13, *Bx7/10–12/14*, and *Bx8/9* genes in green plants are intronless genes, and most *Bx1* genes in green plants are intron-rich genes. In longan, *DlBx1* exhibited the highest count of introns, totaling eight introns. *DlBx2–4*, *DlBx7*, and *DlBx8/9* each contained one intron. The majority of DlBx6 genes contained two introns, while *DlBx6e* was the only one with four introns. Consequently, with the exception of *DlBx1* and *DlBx6e*, the remaining *DlBx* genes were devoid of introns.

### 2.3. Chromosomal Distribution, Gene Duplication, and Synteny Analysis of Benzoxazinoids Biosynthesis Genes in Longan

According to the longan genome annotation, 27 *DlBx* genes were distributed in nine chromosomes (Chr01, Chr04, Chr05, Chr07, Chr08, Chr09, Chr13, Chr14 and Chr15) (Figure 3), and the number of *DlBx* genes from chromosome was 4, 3, 3, 1, 7, 2, 2, 2 and 3, respectively, in which chromosome eight has the maximum number and chromosome seven has the minimum number. We used One Step MCScanX of TBtools-II v2.080 to explore duplications within the longan *DlBx* gene family. The results showed that there were 18 tandem duplication (TD) genes and four segmental or whole-genome duplications (S/WGD) genes in *DlBx* family members, among which one pair of segmental duplications genes was *DlBx6a/DlBx6d* (Appendix A). According to Jian Cheng [31], the *DlBx* family has five TD gene pairs, namely *DlBx3b/DlBx3c*, *DlBx3d/DlBx3e*, *DlBx3f/DlBx3g*, *DlBx6b/DlBx6c* and *DlBx6f/DlBx6g*. Hence, it is possible that TD events play a significant role in driving the amplification of the *DlBx* gene.

To further study the evolutionary mechanism of benzoxazinoids biosynthesis genes in longan, we constructed a collinear map of longan and ten representative species, including Algae (*Chlamydomonas reinhardtii*), bryophyta (*Physcomitrium patens*), Gymnospermae (*P. abies*), pteridophyta (*S. moellendorffii*), two monocotyledons (*B. distachyon* and *Z. mays*) and four dicotyledons (*Amborella trichopoda*, *Nephelium lappaceum*, *Litchi chinensis* and *D. longan* (JDB)) (Figure 4). *DlBx* did not exhibit a collinear relationship with algae and gymnosperms, so only the collinearity of longan and the remaining species was displayed. *P. patens* had 27 chromosomes and 77 scaffolds, *S. moellendorffii* had 370 scaffolds, and *A. trichopoda* had 614 scaffolds; 27 chromosomes, 27 scaffolds and 2 scaffolds were selected from *P. patens*, *S. moellendorffii*, *A. trichopoda* for collinear analysis based on collinearity files. As shown in Figure 4, there were collinearity between 2, 9, 2, 1, 2, 8, 9 and 12 *DlBx* and *P. patens*, *S. moellendorffii*, *B. distachyon*, *Z. mays*, *A. trichopoda*, *N. lappaceum*, *L. chinensis*, and *D. longan* (JDB), respectively, and 9, 43, 3, 2, 3, 10, 12 and 15 orthologous gene pairs with these plants, respectively.

The *Bx1* gene in six species, including *B. distachyon*, *Z. mays*, *A. trichopoda*, *N. lappaceum*, *L. chinensis*, and *D. longan* (JDB), exhibited colinear gene pairs, indicating that these gene pairs may have originated prior to the divergence of monocots and dicots. DlBx2–4 genes also displayed colinearity with four species, namely *P. patens*, *S. moellendorffii*, *N. lappaceum*, and *D. longan* (JDB), with a significant number of colinear gene pairs, including at least six between *DlBx3d* and *P. patens*, and *S. moellendorffii*. The *DlBx6* genes showed a colinearity with *S. moellendorffii*, *B. distachyon*, *A. trichopoda*, *N. lappaceum*, *L. chinensis*, and *D. longan* (JDB), while *DlBx6a* exhibited a colinearity with these species and *DlBx6c* only displayed colinear gene pairs with *L. chinensis* and *D. longan* (JDB). The *DlBx7* genes exhibited a colinearity with *S. moellendorffii*, *N. lappaceum*, *L. chinensis*, and *D. longan* (JDB), and *DlBx7d* displayed a colinearity with these four species. The *DlBx8/9* genes showed a colinearity with *P. patens*, *S. moellendorffii*, *N. lappaceum*, and *L. chinensis*, with at least five colinear gene pairs between *DlBx9b* and *S. moellendorffii*. Therefore, it was suggested that *DlBx3d*, *DlBx6a*, *DlBx7d*, and *DlBx9b* may play a crucial role in the evolutionary process. Furthermore, only *DlBx3d*, *DlBx9a*, and *P. patens* exhibited colinearity, indicating that the homologous gene pairs formed by these genes may have evolved earliest.

To further understand the role of assessing evolutionary constraints, we calculated the Ks values, Ka values, and Ka/Ks ratios of *Bx* orthologous and paralogous genes to assess the natural selection pressure acting on these genes during evolution (Appendix A). The results showed that only *DlBx3-1/D.long020347.01* had a Ka/Ks value greater than 1, with a value of 1.15543146009824. The Ka/Ks values of the other paralogous and orthologous genes were less than 1, ranging from 0.087874918 to 0.846186725, indicating that the DlBx gene family experienced a strong purifying selection throughout its evolutionary history.

### 2.4. Protein Interaction Prediction, Cis-Acting Elements and Transcription Factor Regulatory Networks Analysis of Benzoxazinoids Biosynthesis Genes in Longan

Using *Z. mays* as a model plant, the String v12.0 [32] protein interaction online database was used to predict the functional relationship between DlBx protein members (Figure 5A). The results showed that DlBx proteins could interact not only with their own family members, but also with other proteins, mainly with indole-3-glycerol phosphate synthase (B4FS35_MAIZE, C0P9Q8_MAIZE and C0PGV5_MAIZE), tryptophan synthase (K7TR93_MAIZE, K7TTR0_MAIZE, A0A1D6P4E8, TSB1 and TSB2), Bf1 and N-(5′-phosphoribosyl) anthranilate isomerase (B4FZQ3_MAIZE and K7V589_MAIZE).

In total, 90 cis-acting elements were found in the promoter region of the benzoxazinoids biosynthesis-related genes in longan through analyzing 2 KB promoter sequences upstream of the *DlBx* initiation codon coding sequence (ATG) (Figure 5B). The promoter regions of the *DlBx* were found to contain abundant cis-acting elements, falling into seven classes as follows: promoter-related, light response, hormone response, developmental-related, environmental-stress-related, site-binding-related, and other elements (Appendix A). The promoter regions of all *DlBx* genes contained a TATA-box and CAAT-box, suggesting that the *DlBx* gene has normal transcriptional levels. In addition, the number of light responsive elements in the *DlBx* promoter region were the largest, followed by hormone responsive elements and environmental stress-related responsive elements, and the number of responsive elements related to development and site-binding was less. Among the light responsive elements, all *DlBx* genes contained Box 4, and at least 52% of the *DlBx* genes contained five elements: G-box (24), GATA-motif (18), TCT-motif (20), and GT1-motif (14). Among the hormone response elements, 89% of the *DlBx* genes contained abscisic acid (ABA) response element ABRE (24), followed by methyl jasmonate (MeJA) response element CGTCA-Motif (17) and TGACG-motif (17). Among the response elements related to environmental stress, the anaerobic-induced response element ARE(18) was widely distributed in the promoter sequence of *DlBx*, followed by the wound-responsive element WUN-motif (13). Among them, the *DlBx1* promoter region contained light-responsive elements (Box 4, G-box and GATA-motif), hormone-responsive elements, namely MeJA (CGTCA-Motif and TGACG-motif) and ABA (ABRE) responsive elements, and low-temperature responsive elements (LTR). Most of the promoter regions of the *DlBx2–4* genes contained these seven types of cis-acting elements, and five light-responsive elements, Box 4 (10), G-box (8), GATA-motif (7), TCT-motif (7) and GT1-motif (7), were widely distributed in the *DlBx2–4* promoter region; 70% of *DlBx2–4* members contained ABA (ABRE) response elements, 50% of *DlBx2–4* members contained auxin-related response elements, and 40% or more *DlBx2–4* contained the following four environmental stress-related response elements: ARE, WUN-motif, LTR, TC-rich repeats and AT-rich element, and two developmental-related response elements, O2-site and CAT-box. The light-responsive elements of Box 4, G-box, and the TCT-motif were widely distributed in the promoter region of the *DlBx6* genes, and *DlBx6* also contained a large number of hormone response elements, namely ABRE, CGTCA-motif, TGACG-motif, and salicylic acid (SA) response element (TCA-element). Most *DlBx6* genes also contained ARE, a drought stress response element (MBS), and a zein metabolism regulation response element (O2-site). The promoter region of the *DlBx7* gene contained three light-responsive elements (Box 4, G-box and GATA-motif), hormone-related ABRE and environmental stress-related ARE, and most of the *DlBx7* genes also contained a light-responsive TCT-motif and GT1-motif, a hormone-related CGTCA-Motif and TGACG-motif, an auxin-response element (AuxRR-Core or TGA-element) and a gibberellin (GA) response element (GARE-motif or TATC-box), a growth and development-related palisade mesophyll cells differentiation element (HD-zip 1), and an environmental stress-related WUN-motif. All *DlBx8*/9 genes contained four response elements, namely Box 4, G-box, TCT-motif, and ABRE. And both *DlBx8* and *DlBx9b* contained GT1-motif, CGTCA-Motif, TGACG-motif, and WUN-motif response elements. *DlBx8* and *DlBx9a* both contained ARE and LTR response elements. *DlBx9a* and *DlBx9b* both contained auxin response elements TGA-element.

To understand the regulatory network of *DlBx* genes by transcription factors, *Arabidopsis* transcription factors were used as bait to predict the transcription factor regulatory network by JASPR2020 using the upstream 2 KB sequence of *DlBx* family members as transcription factor binding regions (relative profile score threshold = 95%) (Figure 5C,D). Our findings revealed that 27 *DlBx* genes were identified as targets of 593 TFs belonging to 19 families. The all *DlBx* genes exhibited a high frequency of presence of Myb-related, HD-ZIP, and PLINC. In addition, the 19 families could be divided into 13 classes, among which homeodomain factors have the largest number, followed by tryptophan cluster factors. Previous studies have shown that the BXs biosynthesis pathway is a branch of the tryptophan biosynthesis pathway, suggesting that tryptophan cluster factors can mediate the expression of genes related to BXs biosynthesis pathway.

### 2.5. Expression Profiling of DlBx Genes during Early Somatic Embryogenesis and Hormone Treatment

In order to understand the expression patterns of *DlBx* in the early SE stages (EC, ICpEC, and GE) of longan, the expression of *DlBx* was analyzed according to the longan transcriptome database. The FPKM values of 22 *DlBx* genes were detected in 27 *DlBx*, and the expression heat maps were prepared (Figure 6A). The results showed that the expression of *DlBx1* was higher in all stages of early SE, and its expression level was the highest in the EC stage. Its expression decreased gradually with early SE. Most of the *DlBx3–4* was expressed at low levels in the early SE stages, and only *DlBx3g* was highly expressed in the ICpEC and GE stages. It was particularly highly expressed in the GE stage. Most of the *DlBx6* genes were expressed in the three stages of early somatic embryogenesis. Among them, *DlBx6b* was expressed at higher levels in all three stages. *DlBx6c* and *DlBx6h* were up-regulated in ICpEC and GE and were highly expressed in GE. *DlBx6a* and *DlBx6d* were expressed at lower or no levels during the early SE process. In contrast, all *DlBx7* were expressed at all stages, but the expression levels were low. The expression of *DlBx7a* and *DlBx7d* showed a gradual downward trend in the early SE process. The gene expression of *DlBx7d* especially was significantly down-regulated in the GE stage. The expression trend of *DlBx8* and *DlBx9b* was the same in the early SE stage, and the expression level was gradually up-regulated. However, *DlBx9a* was not expressed in the EC stage, and its expression in the ICpEC and GE stages were lower. In conclusion, there were significant differences in the expression levels of *DlBx* in the early SE stage of longan. Overall, the expression of eight *DlBx* genes in the early SE stage of longan showed a gradual upward trend, while the expression trend of six *DlBx* genes was reversed. Hormones play an important role in the somatic embryogenesis of plants, and 2,4-D is the key factor affecting the development of somatic embryos in longan. The RNA-seq expression analysis of longan EC at 1 mg/L 2,4-D, 1 mg/L 2,4-D and 5 mg/L Kinetin (KT), 5 mg/L KT, MS treatment showed that 21 *DlBx* genes were expressed in response to different hormonal treatments (Figure 6B). Compared with CK (MS), the expression of 14 *DlBx* genes were up-regulated under 2,4-D treatment, and the expression of five genes (*DlBx3a*, *DlBx3b*, *DlBx3g*, *DlBx4b*, and *DlBx9a*) were up-regulated by more than 2-fold; the expression of 15 *DlBx* genes were up-regulated under 2,4-D+KT treatment; the expression of 13 *DlBx* genes were slightly up-regulated under KT treatment. The results showed that the expression of most *DlBx* genes were up-regulated under treatment containing 2,4-D (2,4-D and 2,4-D+KT), indicating that most *DlBx* genes responded to auxins.

### 2.6. Expression Profiling of DlBx Genes under Light, Temperatures, and PEG Treatments

According to the above, a number of core promoter elements were identified in the promoter sequences of *DlBx*. These elements were involved in light responsiveness, stress responsiveness, and hormonal responsiveness. In this study, transcriptome data were used to analyze the expression patterns of the *DlBx* gene under different light, temperature, and PEG to simulate drought stress. Previous studies in our laboratory have shown that the BXs biosynthesis pathway is significantly enriched under light treatment. Therefore this pathway may play an important role in the light response process [28]. The RNA-seq expression analysis of longan EC treated with blue, dark, and white light showed that 19 *DlBx* genes were expressed in response to light treatments (Figure 6C). The results showed that *DlBx1* was highly expressed under different light quality treatments. The expression levels of *DlBx3b* and *DlBx4b* were lower or no different under light quality treatments, the expression of *DlBx3g* was higher under dark treatment, and the expression was significantly down-regulated under blue light and white light treatment. In *DlBx6*, the FPKM values of other genes except *DlBx6* could be detected, and there were three main expression patterns. The first was that the expression of *DlBx6a* and *DlBx6e* was up-regulated under dark conditions and down-regulated under blue light conditions. The second was *DlBx6b*, *DlBx6c*, and *DlBx6h*, which were down-regulated in dark conditions and up-regulated in blue light conditions. The third was that the gene expression levels of *DlBx6g* and *DlBx6f* were up-regulated under dark and white light conditions, but down-regulated under blue light conditions. In addition, *DlBx6f* and *DlBx6h* were differentially expressed genes. Therefore, it is suggested that *DlBX6f* and *DlBX6h* are key genes in the longan BXs biosynthesis pathway. The expression of *DlBx7* was low under different light quality treatments, and the expression of the *DlBx7d* gene was significantly down-regulated under blue light treatment. Compared to the dark treatment, *DlBx8*, *DlBx9a*, and *DlBx9b* were up-regulated under blue and white light treatments.

Based on the transcriptome data under different temperature treatments, it was found that 23 *DlBxs* were expressed in response to temperature treatment under the conditions of longan EC treatment at 15 °C, 25 °C, and 35 °C (Figure 6D). The results show that, ten and three *DlBx* genes were upregulated under 35 ℃ and 15 ℃, respectively, compared to 25 ℃. It can be seen that most *DlBx* responds to temperature stress. In our laboratory, 5% PEG and 7.5% PEG were used to treat longan EC to simulate drought stress (Figure 6E). According to the transcriptome data, 25 *DlBxs* responded to drought stress. However, most of the *DlBxs* were expressed at low levels under drought stress. Compared to CK, six and seven *DlBxs* were upregulated under the 7.5% PEG and 5% PEG, respectively.

### 2.7. Responses of DlBx Genes in Longan to Exogenous IAA and DIMBOA

To understand the possible function of *DlBx* in auxin, we analyzed the expression profile of key members of *DlBxs* in longan callus treated with different concentrations of exogenous IAA. As shown in Figure 7A, the expression of *DlBx1* was significantly up-regulated under 1.0 mg/L IAA treatment, and down-regulated at other concentrations, especially under 2.0 mg/L IAA treatment. The expression of *DlBx6f* was significantly down-regulated under 1.5 mg/L IAA treatment. The expression of *DlBx6h* was significantly up-regulated under different concentrations of IAA, and the expression level was the highest at 1.0 mg/L IAA treatment. Thus, this suggests that *DlBx* responds to synthetic auxin.

In order to understand the role of DIMBOA in early SE of longan, we analyzed its effect on the morphogenesis of early SE and the expression profile of *DlBx* under DIMBOA treatment. The results showed (Figure 7B) that at the seventh day, compared with CK, the longan embryonic cells treated with different concentrations of DIMBOA showed a more compact state, especially under 0.1 mg/L DIMBOA treatment. At the ninth day, CK was still in the pre-GE stage, but under the 0.1 mg/L and 1.0 mg/L DIMBOA treatments, the longan embryonic cells had formed an early GE structure, and especially under the 0.1 mg/L DIMBOA treatment, the longan embryonic cells were more compact. On the 11th day, the GE appeared and the cell edges were smoother in the treatment of more than 0.1 mg/L DIMBOA compared with CK. After 13 days of treatment, typical GE appeared in both the control group and the treatment group. The size of GE in longan was generally greater than that of CK under 0.1 mg/L DIMBOA treatment. Cell morphology after treatment with other DIMBOA concentrations were presented in Appendix A. Further, we analyzed the expression of eight *DlBxs* (expression was significantly up- or down-regulated in the GE) under different concentrations of exogenous DIMBOA at 9 and 13 days (Figure 7C). The results showed that, compared with CK, at the ninth day the expression of *DlBx6h*, *DlBx8* and *DlBx9b* were down-regulated under 1.0 mg/L DIMBOA treatment for nine days, in which the expression of *DlBx8* and *DlBx9b* were significantly down-regulated, and the expression of other genes were significantly up-regulated. The situation was reversed at the 13^th^ day; the expression of *DlBx1* and *DlBx7d* were up-regulated with a slight change in the expression level of *DlBx3g*, and the expression level of the other genes showed a significant downward trend, particularly under treatment with 1.0 mg/L DIMBOA. In general, the expression level of DlBxs showed an upward trend under DIMBOA treatment at the ninth day, and down-regulated expression levels of *DlBx* gene under 1.0 mg/L DIMBOA treatment at the thirteenth day. These results indicated that DIMBOA treatment could promote SE and affect the expression level of *DlBxs*.

In order to understand the effect of DIMBOA on endogenous auxin, this study examined the content of endogenous IAA in embryogenic longan cultures under different concentrations of DIMBOA treatment (Figure 7D). The results showed that compared to CK, the endogenous IAA content showed an upward trend at the seventh and ninth day after treatment with different concentrations of DIMBOA. Furthermore, the endogenous IAA content increased significantly under low concentration DIMBOA treatment at the seventh day. On the 11th and 13rd day, the endogenous IAA content of longan embryonic cultures showed a decreasing trend after different concentrations of DIMBOA treatment. The endogenous IAA content decreased significantly under high concentration DIMBOA treatment on the 13rd day. In conclusion, 0.1 mg/L DIMBOA had the most significant effect on the accumulation of endogenous IAA content, with peaks at the 7th and 13th day of treatment, respectively. Its change trend showed the letter “U”. It can be seen that DIMBOA affected the accumulation of endogenous IAA content in longan embryonic cultures. In order to further understand whether *DlBxs* are related to IAA content, the correlation between the expression levels of *DlBx* and IAA content after 9 and 13 days of DIMBOA treatment was analyzed. The results show (Figure 7E) that *DlBx* gene expression was negatively correlated with IAA content, while *DlBx* gene expression was significantly positively correlated with *DlBx* gene expression. It is worth noting that the IAA content was significantly negatively correlated with the *DlBx1* gene expression level, and the expression level of *DlBx6h* was negatively correlated with the expression level of *DlBx7d*.

## 3. Discussion

### 3.1. The Bx Genes Evolved Independently in Monocots and Dicots

Studies have shown that BXs are widely found in monocots and less commonly in dicots. Multiple *Bx* genes responsible for controlling BXs biosynthesis have been identified, namely 14 genes in maize (*ZmBx1–ZmBx14*) [33], 10 genes in wheat (*TaBx1–TaBx5*, *TaGTa–TaGTd*) [34], 5 genes in *H. Lechleri* (*HlBx1-HlBx5*) [35], 8 genes in rye (*ScBx1–ScBx7*) [34,36,37], etc. In this study, a total of 27 members (*DlBx1*, *DlBx2*, *DlBx3a–g*, *DlBx4a–b*, *DlBx6a–h*, *DlBx7a–e*, *DlBx8*, *DlBx9a–b*) were identified in longan, a larger number compared to grasses. To further understand benzoxazinoids biosynthesis in longan, we identified a total of 27 *DlBx* genes in its genome and performed phylogenetic analyses using genes from 24 species, including algae, bryophyta, pteridophyta, gymnospermae, monocots and dicots.

In the evolutionary origin of the *Bx* genes in gramineous plants, Dongya Wu et al. [38] demonstrated that the *Bx* genes in Triticeae originated from Panicoideae via horizontal transfer (HT). In this study, the homologous genes of *Bx1* had originated in Algae, while the homologous *2-ODD* and *UGTs* genes for benzoxazinoid biosynthesis were found only in angiosperms. This suggests that the benzoxazinoids biosynthesis pathway may have emerged after the differentiation of monocots and dicots. Furthermore, the core *Bx* gene responsible for the synthesis of BXs was observed to form biosynthetic gene clusters in multiple species in gramineous plants. For example, the *ZmBx1–ZmBx8* were located on chromosome 4 in *Z. mays* [39]. In rye, the *ScBx1* and *ScBx2* were located on chromosome 7, while the *ScBx3–ScBx5* were located on chromosome 5, and a similar situation existed in wheat; *TaBx1* and *TaBx2* were located on chromosome 7R, while *TaBx3–TaBx5* were located on chromosome 5R, indicating that the Bx cluster underwent gene rearrangement during evolution [40]. Previous studies proposed that *Bx1* evolved via gene duplication and neofunctionalization from the TSA gene [11,35]. *Bx1* was of monophyletic origin in monocots and evolved independently in monocots and dicots [41]. In longan, the type III of the *DlSAMS* gene also evolved independently in monocots and dicots [42]. In *Z. mays*, *Bx3*, *Bx4*, and *Bx5* are the result of tandem duplication of *CYP71C* and form a gene cluster with *Bx2* [13]. Previous studies have shown that *Bx2–5* was structurally highly homologous, but the protein encoded by *Bx2–5* was highly substrate-specific, suggesting that this may be due to the new functionalization of a common ancestor gene after replication [43]. The second step of the pathway leading to the production of indolin-2-one from indole was a P450-dependent activity in the DIBOA-producing dicots [2] as in Poaceae, but as no *CYP71C* genes were found in dicots; therefore, the P450 enzyme involved in the BXs biosynthesis pathway of dicots was not an orthologous gene, but may have recruited paralogous genes to participate in the pathway [43]. In this study, there were three pairs of TD genes in longan *Bx3*, but they were distributed on different chromosomes. This suggests that there may have been loss or re-duplication after gene rearrangement during the evolution of dicots. The dicot *CoBx8* and the *Bx8* enzymes from grasses were found in distinct groups, and both groups separated before the diversification of dicots. The phylogeny indicates independent evolution of UGTs for benzoxazinoid biosynthesis in monocots and dicots [44]. In this study, based on the phylogenetic classification of *Bx* protein in green plants, it was found that five gene families related to BXs biosynthesis were clearly distinguished in monocots and dicots. This further verified that *Bx* genes evolved independently in monocots and dicots.

Gene duplication is a major mechanism for the generation of new genes. S/WGD and TD are the main ways in which gene duplication occurs. After gene duplication, some duplicated genes undergo neofunctionalization, whereas others maintain largely redundant functions [45]. Duplicated genes exhibit various degrees of functional diversification in plants [46]. Monocots and dicots were estimated to diverge between 171 and 203 million years ago (MYa) [47]. Longan diverged about 69.3 MYa [48]. With the advent of monocots and dicots, the *Bx* genes were differentiated, especially CYP71, 2-ODD, OMT, and UGTs for benzoxazinoid biosynthesis to produce new genes through replication. Four duplication forms were detected in *DlBx*, including singleton (1), proximal (4), TD (18), and S/WGD (4). It is clear that the expansion of the *DlBx* gene family in longan is related to TD and S/WGD, among which TD is the main driving force of the evolution of longan *DlBx*, and longan *DlBx* is strongly constrained by purification selection during the evolutionary process.

### 3.2. DlBx May Be Involved in the Hormonal Response and Stress Response Process of Early Somatic Embryogenesis in Longan

Hormones play an important role in the plant SE process. To date, BXs have been extensively studied and shown to regulate signaling pathways such as auxin, cytokinin, and GA. Cis regulatory elements are typically non-coding DNA that contain binding sites for regulatory molecules (e.g., transcription factors) required to activate and maintain transcription [49]. Studies have shown that jasmonate (JA) increases the concentration of DIMBOA-Glc in wheat seedlings and upregulates the expression of *TaBx1–TaBx6* genes [50]. Zhang [51] et al. showed that *ZmMPK6* and ethylene signaling specifically and commonly regulate the transcription of other benzoxazinoid biosynthetic genes. BXs also inhibit GA-induced α-amylase activity, which regulates GA signaling. This in turn regulates stem elongation [52]. In addition, the benzoxazine biosynthesis pathway was the only enrichment pathway in the *qd* (‘*quick development*’ mutant) mutant plants of barley, and *Bx3*, *Bx4*, *Bx5*, and *Bx8–9* were down-regulated [53]. SA increased the endogenous DIMBOA content in plants [54]. The study of the cis-acting elements of the promoter of the *DlBx* gene revealed that the promoter region of *DlBx* contains MeJA, Auxin, SA, and GA response elements, suggesting that the *DlBx* gene may be involved in the early somatic embryogenesis hormone response process of longan. Longan transcriptome data showed that the expression levels of most *DlBx* genes were significantly increased under 2,4-D treatment and 2,4-D+KT. In addition, the expression levels of *DlBx1*, DlBx6f and *DlBx6h* changed significantly under different IAA concentrations, indicating the crucial role of *DlBx* in the hormone response of longan, especially auxin.

A large number of studies have shown that the concentration and expression of synthesis-related genes in plants are significantly affected by growth environment conditions, and BXs participate in various defense responses to help plants resist biotic and abiotic stresses. For example, the concentration of DIMBOA in maize leaves was low under strong light and UV conditions [55,56]. In wheat, the DIMBOA-Glc concentration of seedlings was negatively correlated with light intensity [57]. In longan, the BXs synthesis pathway was enriched under different light quality treatments. *TaBx* genes were highly induced under drought treatments (*TaBx1* and *TaBx4*), PEG treatments (most *TaBx* genes), and cold stress (*TaBx3*, *TaBx4* and *TaBx8/9*) in wheat [58]. Under low temperature treatment (4 °C), the concentration of BXs and the expression of the *Bx* gene in rye plants showed a decreasing trend [59]. In wheat, the cis-acting elements MBS, G-box, and GAG-motif were found in all *TaBx3* and *TaBx4* homoeologs and orthologs. MBS was associated with drought responsiveness, while the G-box and GAG-motif were linked to light responsiveness. These three cis-acting elements were also present in *ZmBx4*. The phase-specific transcription of *TaBx3* and *TaBx4* has been confirmed to be controlled by some important cis-elements [60]. In this study, the *DlBx* promoter region also has abiotic stress response elements such as low temperature, drought, anaerobic induction, wound, and anoxic, suggesting that *DlBx* gene may be involved in the process of stress.

### 3.3. DIMBOA Might Promote Early Somatic Embryogenesis in Longan through Affecting Auxin Synthesis

Previous studies have shown that hormones play a vital role in the regulation of SE, especially auxin and cytokinin [61]. BXs and their breakdown products have been associated with the regulation of auxin signaling. DIBOA and BOA inhibit the binding of 1-naphthylacetic acid (NAA) to auxin receptors, as well as auxin-induced coleoptile growth in maize [62]. Subsequent experiments with oat, timothy grass, amaranth, and pea showed that the exogenous addition of DIMBOA and/or MBOA affects auxin-induced growth [63,64,65]. Early studies found that the effect of DIMBOA on oat roots was concentration-dependent, with DIMBOA concentrations less than 1.5 mM promoting root growth and more than 1.5 mM inhibiting root growth [66]. A total of 0.6 mM DIMBOA promoted alfalfa seed germination and seedling growth, and DIMBOA can effectively improve the tolerance of alfalfa seedlings to coumarin stress [67]. DIMBOA, a main hydroxamic acid, commonly exists in the form of DIMBOA glucoside [68]; it has been detected in callus from wheat, barley, and maize [69,70,71]. Among the barley callus, the content of DIMBOA-Glc in the renewable callus was higher than that in the non-renewable callus, and the regeneration ability of the callus was closely related to DIMBOA-Glc [69]. Xinyi He [69] et al. argued that DIMBOA and its glucoside may indirectly affect the regeneration capacity of callus by altering other components, especially phytohormones. Through microscopic observation, DIMBOA promoted the early SE of longan. *DlBx3g*, *DlBx6c*, *DlBx6h*, *DlBx8*, *DlBx9b*, *DlBx1* and *DlBx7d* played an important role in the early stage of longan SE and had diverse expression patterns under different concentrations of DIMBOA, which suggested that these *DlBx* genes play an important role in the promotion of early SE in longan by DIMBOA. The BXs biological synthesis pathway and the auxin biological synthesis pathway share the same precursors, and the content of endogenous IAA directly affects the further development and maturation of longan somatic embryos [24]. Therefore, in order to explore whether the effect of DIMBOA on the somatic embryogenesis of longan is related to IAA, the endogenous IAA content of longan embryonic cultures treated with exogenous DIMBOA was determined. The results showed that DIMBOA significantly affected the accumulation of endogenous IAA in longan embryogenic culture. Early studies have shown that two stages of EC and GE were the key turning points in the process of longan SE, and the endogenous IAA content peaked in the two stages of EC II and GE, and the endogenous IAA content was significantly lowered after GE, thereby promoting the further development and maturation of longan somatic embryo [24]. There were multiple enzymes that could convert IGP to indole in maize, but only one gene *DlBx1* was identified in longan, so *DlBx1* played an important role in both auxin and BXs biological synthesis pathways. *DlBx1* expression was significantly up-regulated in the early stage of DIMBOA treatment, and the expression level of *DlBx1* was significantly negatively correlated with the endogenous IAA content. In addition, IAA content increased significantly and decreased significantly in the early stage (seventh day) and late stage (thirteenth day) of DIMBOA treatment, respectively. Therefore, in the early stage of longan embryonic cultures treated with DIMBOA, DIMBOA might inhibit the biological synthesis of BXs compounds and thus promoted the biological synthesis of IAA by mediating the expression of *DlBx1* through negative feedback (Figure 8). In addition, previous studies have shown that exogenous DIMBOA treatment could promoted the biological synthesis of H_2_O_2_ [64]. In recent years, it has been found that H_2_O_2_ could regulated the accumulation and redistribution of auxin by regulating the polar transport of auxin [72], and mainly regulated the growth and development of plants by inhibiting auxin signaling in plants [73]. Therefore, in the late stage of longan embryonic cultures treated with DIMBOA, DIMBOA reduced the IAA content, which may be caused by promoting H_2_O_2_ accumulation, thereby inhibiting auxin biological synthesis and transportation (Figure 8). These results suggest that DIMBOA might indirectly promote longan SE by regulating the content change in endogenous IAA via affecting the expression level of *DlBx1* and the content of H_2_O_2_.

## 4. Materials and Methods

### 4.1. Plant Materials

Taking the ‘HHZ’ longan EC as the primary material, the longan EC material was cultured with reference to Lai Zhongxiong’s culture method [74]. For DIMBOA treatment, EC was transferred to Murashige and Skoog (MS) media after 20 days of proliferation. DIMBOA with concentrations of 0.05 mg/L, 0.1 mg/L, 0.2 mg/L, 0.4 mg/L and 1.0 mg/L were added to treat the materials for 13 days, and the somatic embryo differentiation status was observed using optical microscope at 7, 9, 11, and 13 days. For IAA treatment, the concentration was 0.5 mg/L, 1.0 mg/L, 1.5 mg/L and 2.0 mg/L for 24 h. These materials were quickly transferred to liquid nitrogen and stored in a −80 °C freezer for subsequent RNA extraction and endogenous IAA content determination in a series of experiments. Three independent biological replicates were made.

### 4.2. Identification of Benzoxazinoids Biosynthesis Genes in Longan

The genomic and amino acid sequence of Chlamydomonas reinhardtii, Ostreococcus tauri, Marchanta polymorpha, Physcobmitrium patens, Ceratopteris richardii, Selaginella moellendorffii, Brachypodium distachyon, Oryza sativa, Amborella trichopoda, Citrus sinensis, Malus domestica, Populus trichocarpa, Arabidopsis thaliana, Litchi chinensis, Nephelium lappaceum, Sapindus mukorossi, Picea abies, and Pinus taeda were retrieved from the PGDD (http://chibba.agtec.uga.edu/duplication/, accessed on 11 December 2023), NCBI database (https://www.ncbi.nlm.nih.gov, accessed on 11 December 2023), TAIR database (http://www.arabidopsis.org, accessed on 11 December 2023), SapBase (http://www.sapindaceae.com/about.html, accessed on 11 December 2023) and Plantgenie (https://plantgenie.org/, accessed on 11 December 2023), respectively. The whole genome sequences of longan were obtained from the D. longan library (SRR17675476).

The accession numbers of the Bx amino acid sequences of four species, namely maize (*Zea mays*), rye (*Secale cereale*), wild barley (*Hordeum lechleri*) and common wheat (*Triticum aestivum*), were obtained from the literature and downloaded from NBCI [75]. These *Bx* genes were identified using a bidirectional BLAST method. First, the protein sequences of *Bx* in *Z. mays* were used as bait to search the *Bx* of 24 species including longan using TBtools-II v2.080 software [76], and the homologs of *Bx* gene were screened using parameters with an e-value less than 1e-30 and identity greater than 40%. Then, the NCBI online website was used to verify it via BLASTP. Finally, the candidate sequences were verified using Pfam (http://pfam.xfam.org/, accessed on 15 December 2023) and CDD databases (https://www.ncbi.nlm.nih.gov/cdd/?term=, accessed on 15 December 2023) to see if they contain specific domains to obtain homologous gene candidate sequences of the *Bx* gene. The naming method for the *ZmBx* gene could be used as a reference for the naming of the *Bx* gene family in longan. And according to the location of chromosome, the longan *Bx* were named *DlBx1~9b* in turn. ExPasy tools [77] (http://web.expasy.org/protparam/, accessed on 21 December 2023), TMHMM-2.0 [78] (https://services.healthtech.dtu.dk/services/TMHMM-2.0/, accessed on 21 December 2023), and WoLF PSORT [79] (https://wolfpsort.hgc.jp/, accessed on 21 December 2023) were used to obtain corresponding DlBx protein sequence length, molecular weight, isoelectric point, instability coefficient, hydrophilicity, transmembrane, and subcellular localization, respectively.

### 4.3. Phylogenetic Analysis, Gene Structure, Conserved Domains and Motifs Analysis

Phylogenetic trees were constructed using the Bx protein sequences of longan-HHZ and the other 23 species (*C. reinhardtii*, *O. tauri*, *M. polymorpha*, *P. patens*, *C. richardii*, *S. moellendorffii*, *P. abies*, *P. taeda*, *B. distachyon*, *H. lechleri*, *O. sativa*, *S. cereale*, *T. aestivum*, *Z. mays*, *A. trichopoda*, *A. thaliana*, *C. sinensis*, *D. longan*-JDB, *L. chinensis*, *M. domestica*, *N. lappaceum*, *P. trichocarpa*, *S. mukorossi*). First, the muscle online software was used for evolutionary sequence alignment. Then, a maximum likelihood (ML) phylogenetic tree was built with an IQ-tree tool (bootstrap number set to 1000) and the best substitution model was automatically selected. Finally, the phylogenetic tree was annotated and visualized using iTOL v6, based on the longan genome annotation file to identify exons, introns, and untranslated region (UTR) arrangements of DlBx genes. The conserved domains of the DlBx protein were obtained using the CDD databases. The full-length conserved motif of the DlBx protein was obtained using the online MEME tool (https://meme-suite.org/meme/index.html, accessed on 25 December 2023), and the maximum conserved motif search value was set to 20. The gene structure, conserved domains and conserved motifs were visualized using TBtools-II v2.080 software.

### 4.4. Chromosomal Location, Synteny Analysis and a Non-Synonymous/Synonymous Substitution (Ka/Ks)

The chromosomal location information of the *DlBx* were obtained from the longan genome, and the TBtools was used to generate a chromosome map of the *DlBx*. We used Dual Systeny Plotter software (https://github.com/CJ-Chen/TBtools) to construct a collinear analysis map between the ‘HHZ’ longan and ten species: *C. reinhardtii*), *P. patens*, *P. abies*, *S. moellendorffii*, *B. distachyon* and *Z. mays* and *A. trichopoda*, *N. lappaceum*, *L. chinensis* and *D. longan*-JDB. The simple Ka/Ks calculator in TBtools-II v2.080 software was used to calculate the Ka/Ks value of all orthologous gene pairs to estimate the selection mode of *Bx* between ‘HHZ’ longan and the other species. Ka/Ks < 1 indicates purify selection, Ka/Ks = 1 represents neutral selection, and Ka/Ks > 1 indicates positive selection.

### 4.5. Cis-Acting Elements Analyses, Transcription Factor Regulatory Network Prediction and Protein–Protein Interaction Networks

We used the online software PlantCARE v1 (http://bioinformatics.psb.ugent.be./webtools/plantcare/html/, accessed on 27 December 2023) and JASPR2020 (http://jaspar.genereg.net/, accessed on 28 December 2023) to analyze cis-acting elements and TFs in the 2 kb promoter sequence upstream of the DlBx gene, and they were visualized using TBtools-II v2.080 and Cytoscape software, respectively. We used the online website STRING v12.0 [32] (https://cn.string-db.org/, accessed on 27 December 2023) and maize as a probe sequence to predict DlBx protein–protein interactions, and the confidence level was set to 0.7.

### 4.6. Analysis of Specific Expression of DlBx Genes Family

The RNA-Seq data were used to detect the expression of the *DlBx* gene in the early somatic stage of longan and the response to various treatments. We extracted FPKM values for 27 *DlBx* from the following longan transcriptome databases, i.e., early SE stage (EC, ICpEC, and GE), different light quality treatments (black, blue, and white) [80], different temperature treatments (15 °C, 25 °C, and 35 °C), hormone treatments (2,4-D, 2,4-D+KT, KT, and MS), and PEG treatments (7.5%, 5%, and CK) (Appendix A). The expression level of each gene was calculated as FPKM, transformed as log2 (FPKM + 1) (Appendix A). The heatmap was created using TBtools-II v2.080 and the parameters were as follows: Log Scale = Log2, Row Scale, Scale Method = Normalized, Cluster Rows, and Show Ori Value.

### 4.7. Total RNA Extraction, cDNA Reverse Transcription and qRT-PCR Analysis

The total RNA was extracted from all samples using TransZol Up (TRANS), and cDNA synthesis was carried out according to the instruction manual of the SMARTTM RACE cDNA Amplification Kit TransScript miRNA First-Strand cDNA Synthesis SuperMix. DNAMAN 2.0 was used to design and check the specificity of gene-specific primers (Appendix A). This was amplified with the cDNA 10-fold dilution as a template, and a qRT-PCR detection in a Roche LightCycler 96 instrument was performed. Ubiquitin (UBQ) and EF-1α were used as reference genes for qRT-PCR in DIMBOA and IAA-treated samples, respectively.

### 4.8. Determination of Endogenous IAA Content

The levels of IAA were determined with enzyme-linked immunosorbent assay (ELISA). Briefly, 0.1 g sample was grinded with 1 mL PBS solution and extracted, with 10 μL mixed via sample diluent. The mixtures were placed into wells at 37℃ for 30 min, and were washed 5 times. After the above steps were repeated once more, 50 μL of chromogenic agent A and B was placed into the wells at 37 °C for 10 min. Finally, 50 μL of termination liquid was placed into the wells for the stopping reaction, and the blue liquid turned yellow. A microplate reader (Infinite M 200 PRO, Tecan, Switzerland) was used to determine the absorbance value at a 450 nm wavelength. The Pearson’s correlation coefficient was determined using the IBM SPSS Statistics 26, and TBtools-II v2.080 was used for visualization.

## 5. Conclusions

In this study, we investigated the phylogenetic classification and characterization of five gene families associated with benzoxazinoids biosynthesis in 24 green plants from algae to angiosperms. Based on the *D. longan* genome, we conducted an in-depth analysis of the *DlBx* gene family, and 27 *DlBx* genes were identified. The chromosome distribution, cis-acting elements, transcription factors, expression patterns and the exogenous treatment of *DlBxs* were analyzed. The chromosome distribution, cis-acting elements, transcription factors, and expression patterns of *DlBx* were analyzed, as well as the expression of key genes under different exogenous treatments. The expression profile showed that most of the *DlBx* genes showed a specific expression during the longan early SE, which was widely involved in hormone response and abiotic stress. The exogenous DIMBOA treatment of longan EC may mediate the expression of *DlBx1*, and then regulate the content change in endogenous IAA, thereby promoting longan early SE. In the early stage of exogenous DIMBOA treatment, DIMBOA might promote the biological synthesis of IAA by mediating the expression of *DlBx1* through negative feedback, while in the later stage of treatment, DIMBOA reduced the IAA content, which may be caused by promoting the accumulation of H_2_O_2_ to inhibit the biosynthesis and transportation of auxin, thereby indirectly promoting the early SE in longan (Figure 8).

## Figures and Tables

**Figure 1 plants-13-01373-f001:**
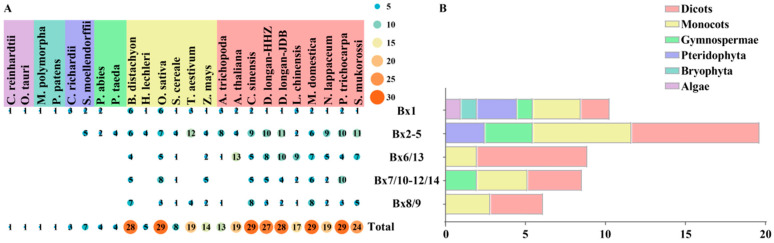
Numbers of *Bx* genes in 24 plant species. (**A**) Numbers of genes in 5 benzoxazinoids biosynthesis gene families from twenty-four plant species. (**B**) The average gene number in each benzoxazinoids biosynthesis gene family in Algae, Bryophyta, Pteridophyta, Gymnospermae, Monocots and Dicots.

**Figure 2 plants-13-01373-f002:**
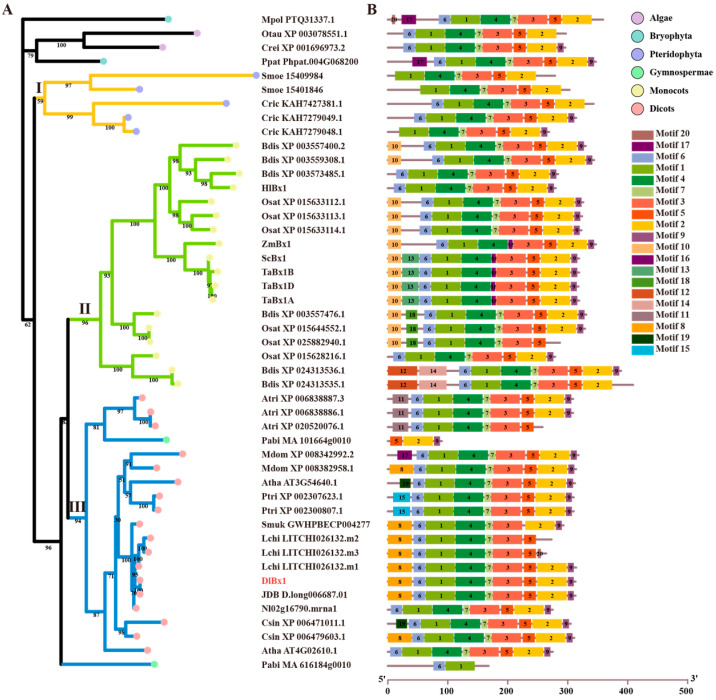
Phylogenetic relationships and conserved motifs of green plant *Bx1* genes. (**A**) A phylogenetic tree of plant *Bx1* genes was built displaying the evolutionary history of these genes. The maximum likelihood (ML) phylogenetic tree was developed using the IQ-TREE tool with 1000 bootstrap replicates. I-III indicates the classification of the phylogenetic tree. (**B**) Motif compositions of plant *Bx1* genes. Differently colored boxes represent different motifs, and the numbers in each box indicates the order of motifs. Motif1: KVAFIPYITAGDPDLSTTAEALKLLDSCGADIIELGVPYSDPLADGPVIQ; Motif2: VTDKP VAVGFGISKPEHVKQIAGWGADGVIIGSAMVKQLGEAASPEEGLK; Motif3: PDLPLEETEALRK EAIKNNJELVLLTTPTTPTERMKAITEASEGFVYLVS; Motif4: AAATRALARGTNLDAVLSMLKE VVPZLSCPIVLFTYYNPILKRGVENFMS; Motif5: GVTGARASVNDRVZTLLQEIK; Motif6: TAAS STVGVSETFSRLKEQG; Motif7: IKDAGVHGLVV; Motif8: MAALKVTTSFLQLKKPDSFSLS RFPSHESNLSIKRFAPMAA; Motif9: LE KFAKSLKSA; Motif10: MAFALKASSSSSSAASSAPASRP; Motif11: WVSPSTPKRSPYFHLSPHSFPSLRVS M; Motif12: MDKEMDARPGTEEARNGWKRARC ASALRFHSDKPSDYTWYVKGRRTKLVD; Motif13: AAVMIPRRRNVLPVIRAVAVAPPAPAPAK; Motif14: DKSLGTLARVTCTETQLSVNEAHGPVAWLGPARESFIVFPKVVHSFC; Motif15: SFLQL KKPETHFJVRNKPPIVSTRRFAPM; Motif16: AGVHGLIV; Motif17: LNHHQN PFCVASCGGARV FSTDKRR; Motif18: PGRRGGAAGRVSFRGVPAPM; Motif19: NRHSHPRDSSLSRKRFT PM; Motif20: CFQNCS.

**Figure 3 plants-13-01373-f003:**
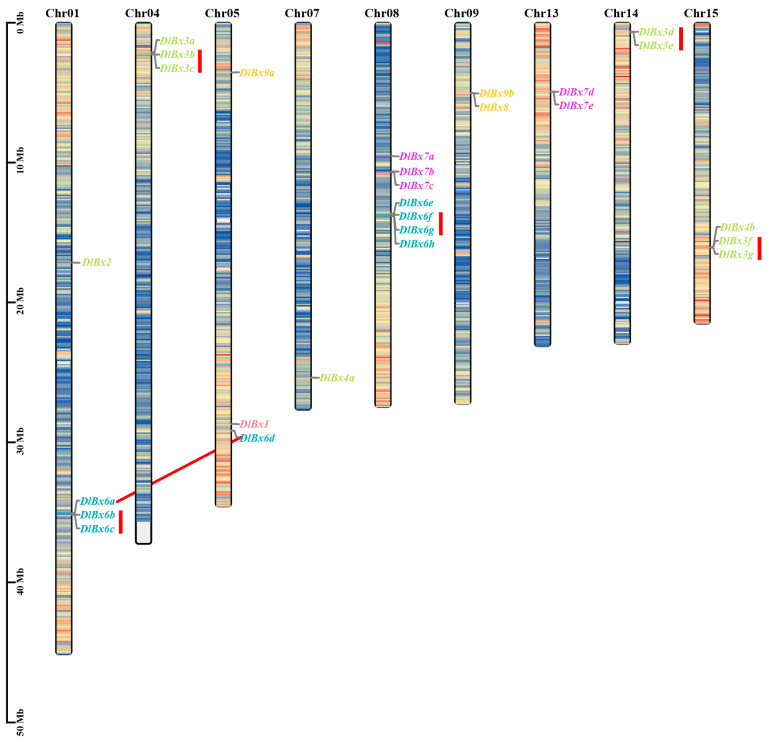
Distribution of benzoxazinoids biosynthetic genes on *Dimocarpus longan* chromosomes. Different gene families are indicated via different colors; red rectangle represents tandem duplicate gene pairs and red lines represent fragment repeat gene pairs.

**Figure 4 plants-13-01373-f004:**
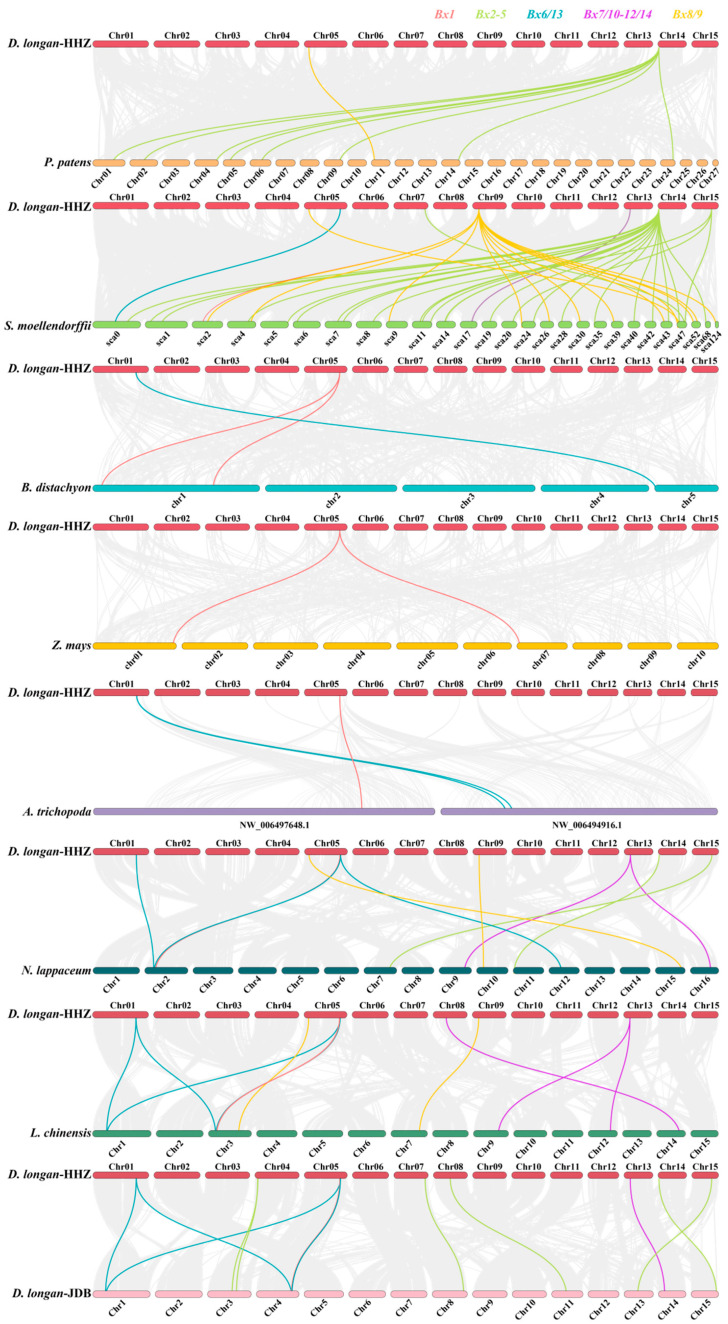
Synteny analysis of *DlBx* genes between longan and other six species. Different colored lines highlight different gene families syntenic gene pairs.

**Figure 5 plants-13-01373-f005:**
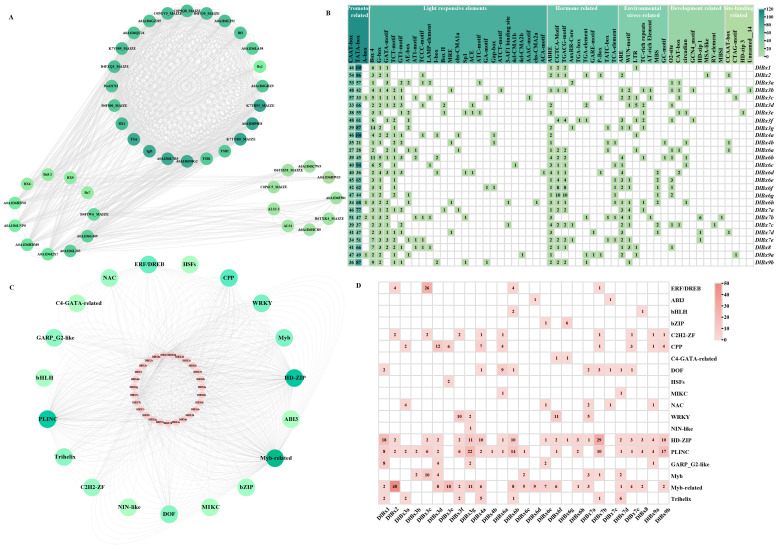
Protein interaction prediction, cis-acting element and transcription factors analysis of benzoxazinoids biosynthesis genes in longan. (**A**) Protein interaction prediction of *DlBx*. (**B**) Cis-acting element analysis of DlBx genes. The 2 kb sequences of 27 *DlBx* genes were analyzed with the PlantCARE v1 software. (**C**) Diagram of transcription factor regulatory network of *DlBx* genes. (**D**) Number of 19 transcription factor families predicted by the *DlBx* genes.

**Figure 6 plants-13-01373-f006:**
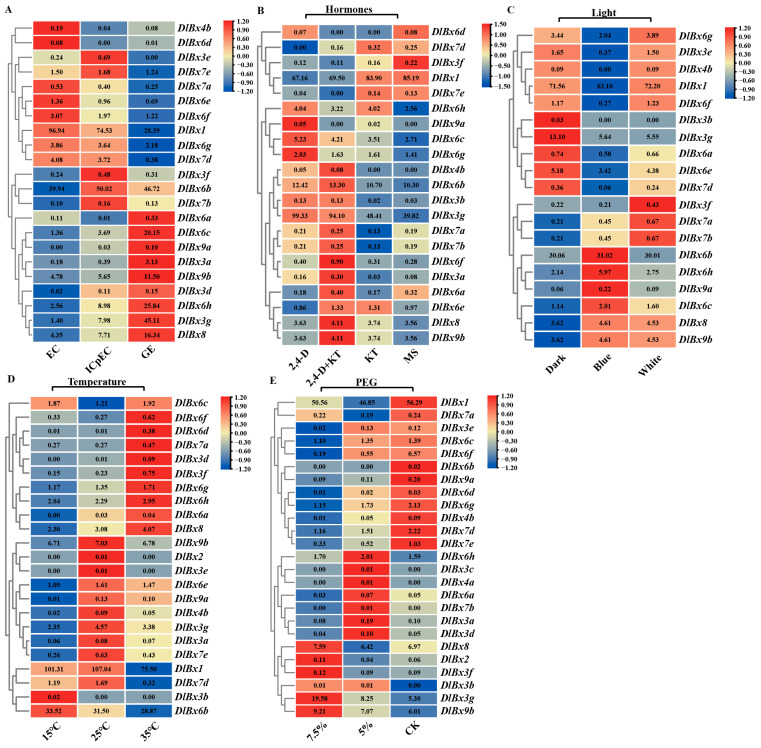
Expression profile and multiomics analysis of *DlBx*. (**A**) 22 *DlBx* with expression detected in RNA-Seq were selected from the EC, ICpEC and GE, (**B**) hormones treatment transcript profiles, (**C**) light treatment transcript profiles, (**D**) temperature treatment transcript profiles, (**E**) PEG treatment transcript profiles. The heat map was produced using TBtools-II v2.080 software, and different colors on the scale bar represent different transcript levels. The data shown in the figure are the original FPKM value, with two decimal places selected.

**Figure 7 plants-13-01373-f007:**
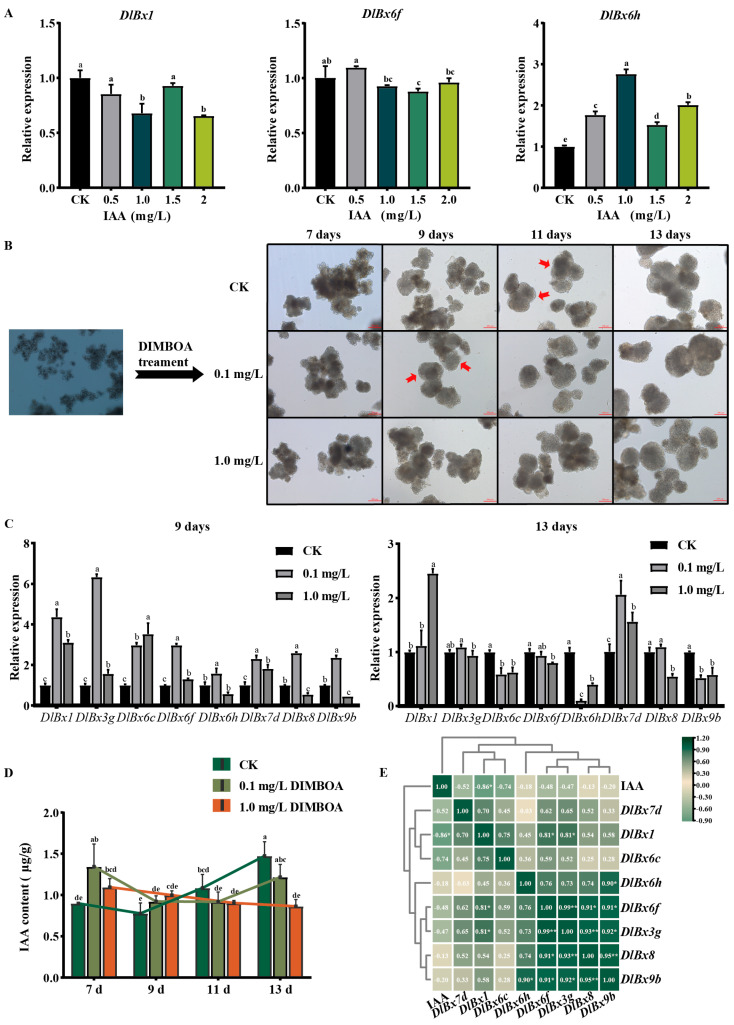
Expression patterns of *DlBx* genes in response to IAA and DIMBOA using qRT-PCR. (**A**) The relative expression of 3 selected *DlBx* genes in response to IAA treatments. The different color blocks represented concentration of treatment. Data were normalized to *EF-1a* gene. (**B**) Morphological change in embryogenic callus under DIMBOA treatment with different concentrations. Scale bar = 200 μm. (**C**) The relative expression of 8 selected *DlBx* genes in different concentrations of DIMBOA treated on different days (9 and 13 d). Data were normalized to *UBQ* gene. (**D**) IAA content in different concentrations of DIMBOA treated on different days (7, 9, 11, and 13 d). CK means control check. Error bars indicate means ± SDs (Duncan’s post hoc test, Different letters indicate the presence of significant, *p* < 0.05). (**E**) Pearson’s correlation matrix of relative expression levels of *DlBx* genes and endogenous IAA content in embryogenic cultures of longan after 9 d and 13 d of treatment with different DIMBOA concentrations. b Positive number: positive correlation; negative number: negative correlation. Asterisks indicate a significant difference (* *p* ≤ 0.05, ** *p* ≤ 0.01).

**Figure 8 plants-13-01373-f008:**
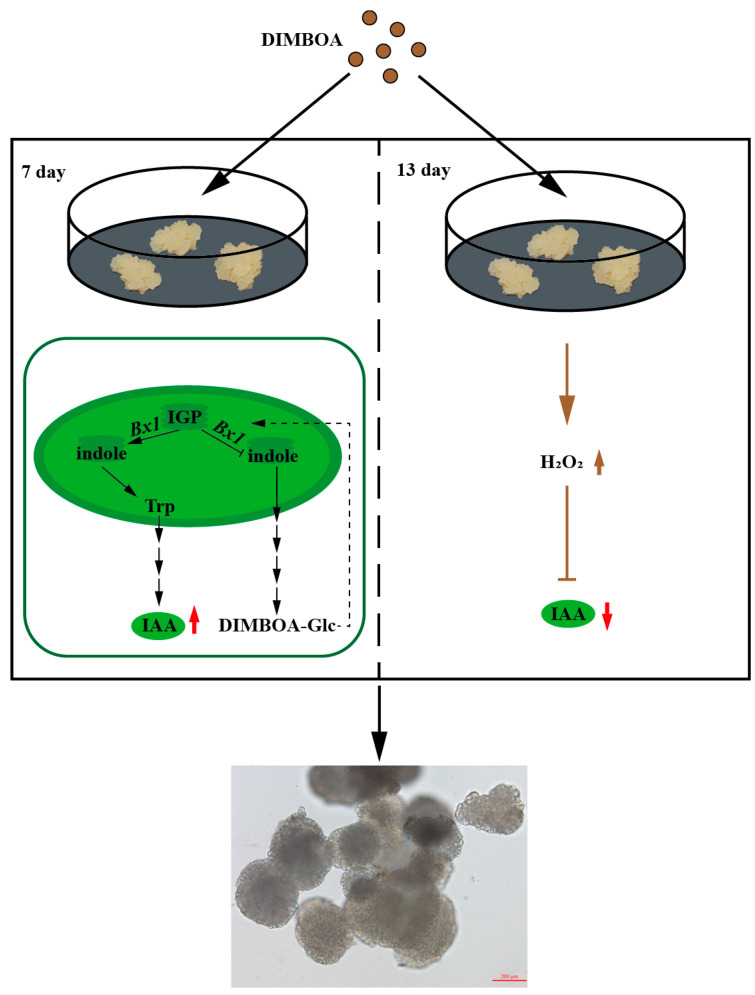
The schematic diagram of DIMBOA promoting early somatic embryogenesis in longan by affecting auxin synthesis. Solid arrows refer to pathways with identified enzymes and dashed arrows refer to undefined pathways. The red arrows indicate the results of this study. Brown arrows indicate the results of previous studies [64,73].

## Data Availability

All relevant data are within the manuscript and Appendix A.

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
