# Peer review of "Genome-Wide Identification and Expression Analysis of *Bx* Involved in Benzoxazinoids Biosynthesis Revealed the Roles of DIMBOA during Early Somatic Embryogenesis in *Dimocarpus longan* Lour"

_plants, 2024, doi:10.3390/plants13101373_

Round 1
Reviewer 1 Report
Comments and Suggestions for Authors
In this manuscript, the presentation of the results as well as their interpretation and discussion is detailed and precise. My only observation/comment is that the materials and methods chapter should be moved immediately after the introduction and before the results.
Author Response
Thank you for your valuable observation/comment. However, the PLANTS journal requires that the materials and methods be placed after the discussion.
Reviewer 2 Report
Comments and Suggestions for Authors
Reviewer comments:
The manuscript ID (plants-2940407) entitled “Genome-wide identification and expression analysis of Bx involved in Benzoxazinoids biosynthesis revealed the roles of DIMBOA during early somatic embryogenesis in Dimocarpus longan Lour” by Xu et al. I found this research topic interesting, demonstrates the evolutionary pattern of Bx genes and regulatory mechanism of DlBx genes during early somatic embryogenesis.
But I have a few concerns related to the research article. I am asking authors to revise the manuscript carefully considering my comments for possible publication in “Plants”.
I have given my comments.
• The present investigation will be a good contribution to the genetic improvement of Sugarcane.
• Line No 136: “Suppl. Table S2”, Table S3, S4 is not found in Suppl. File.
• Line No 584: Authors check and correct “DongyaWuetal. [38]”.
• Line No 674: Authors must check and delete repeated words “process of stress stress”.
• Authors requested to add explant and process to generate EC “longan EC as the primary material”.
• Line number 135: Authors mentioned “Bx genes in each species ranging from 1to 31” but Figure 1 shows 1 to 29 only.
The submitted manuscript may be acceptable for publication after the revision.
Author Response
- Line No 136: “Suppl. Table S2”, Table S3, S4 is not found in Suppl. File.
Thank you for your valuable observation/comment. I've re-uploaded Suppl. Table file.
- Line No 584: Authors check and correct “DongyaWuetal. [38]”.
Thank you for your valuable observation/comment. I've made the check and correct.
- Line No 674: Authors must check and delete repeated words “process of stress stress”.
Thank you for your valuable observation/comment. I've made the check and delete repeated words.
- Authors requested to add explant and process to generate EC “longan EC as the primary material”.
Thank you for your valuable observation/comment. In this experiment, the experimental material longan EC was induced by Zhongxiong Lai, and the explants were taken from the young fruits of longan 40-50 days after flowering, and the young embryos were taken out under sterile conditions for induction culture. For specific cultivation methods, please refer to "Longan Biotechnology Research".
- Line number 135: Authors mentioned “Bx genes in each species ranging from 1to 31” but Figure 1 shows 1 to 29 only.
Thank you for your valuable observation/comment. I've made the check and correct.

Reviewer 3 Report
Comments and Suggestions for Authors
Dear authors, I have reviewed the manuscript entitled:
Genome-wide identification and expression analysis of Bx involved in Benzoxazinoids biosynthesis revealed the roles of DIMBOA during early somatic embryogenesis in Dimocarpus longan Lour.
Authored by:
Xiaoqiong Xu, Chunyu Zhang, Chunwang Lai, ZhiLin Zhang, Jiajia Wu, Qun Su, Yu Gan, Zihao Zhang, Yukun Chen, Rongfang Guo, YuLing Lin, Zhongxiong Lai*
I have tried to review the whole manuscript, but there is a lack of information of abbreviations like: KT, GE, ZI in order to understand what authors mean.
Attached in the pdf manuscript you will find the observation I have found
The references are not well written, there is a mess.
The goal of the manuscript is very good, although the discussion is weak.

Several sentences marked within the pdf manuscript have to be edited.
Author Response
Thank you very much for your input, which has benefited me a lot. In response to your comments, I have made the corresponding changes in the manuscript and made the following explanations based on the questions you raised.
- I have tried to review the whole manuscript, but there is a lack of information of abbreviations like: KT, GE, ZI in order to understand what authors mean.
Thank you for your valuable observation/comment. For the first time in a manuscript, I have added the full name of the abbreviated message to the corresponding place. KT is Kinetin, GE is globular embryo.The abbreviation ZI does not appear in my manuscript.
- The references are not well written, there is a mess.
Thank you for your valuable observation/comment. I have used ednote to reinsert references and have carefully revised the formatting of the references in the manuscript.
- What kind of light enriched BXs?
Thank you for your valuable observation/comment. BXs biosynthesis pathway was significantly enriched under blue light and white light treatment. I have made revisions in the manuscript in the appropriate places.
- Describe this information in material and methods
Thank you for your valuable observation/comment. Methods for predicting protein-protein interactions have been added to Section 4.5 of the manuscript
- What plant species was used to made this networks? Level of confidentiality?
Thank you for your valuable observation/comment. Using JASPR2020 online website, the transcription factor regulatory network predicted by Arabidopsis transcription factors as probe sequences was used. Relative profile score threshold = 95 %. I have made revisions in the manuscript in the appropriate places.
- What type of homeodomain transcription factors? Are they involved in morphology? Do they modulate the embryogenic response?
Thank you for your valuable observation/comment. There are two types of homeodomain transcription factors predicted: HD-ZIP and PLINC. HD-ZIP has the highest number. And a large number of studies have found that HD-ZIP was involved in plant embryogenesis, as could be seen in the review ”Regulatory function of homeodomain-leucine zipper (HD-ZIP) family proteins during embryogenesis”。
- According to the figura 6B, 9 were upregulated while 9 were downregulated and the rest did not change. 2,4-D combined with KT?. maybe Kinetin increase the number of DlBX upregulated to 13. It is the combination of two hormones the best treatment? And at least five does not respomd to hormones and response to white light.
Thank you for your valuable observation/comment. Compared with CK (MS), the expression of 14 DlBx genes was up-regulated under 2,4-D treatment, and the expression of 5 genes (DlBx3a, DlBx3b, DlBx3g, DlBx4b, and DlBx9a) was up-regulated by more than 2-fold; the expression of 15 DlBx genes was up-regulated under 2,4-D+KT treatment; the expression of 13 DlBx genes was slightly up-regulated under KT treatment. Therefore, it was concluded that the effect of 2,-D treatment on DlBx expression was greater than that of KT. In the process of culture and maintenance of longan EC, 2,4-D and KT are essential hormones. Therefore, this study explored the expression pattern of Bx under 2,4-D,2,4-D+KT,KT treatment, in order to explore whether Bx plays a role in the maintenance of longan EC.
- Why did you did not use red light? It is clear that the set of BX that didnt respond to hormones are repressed by blue light.
Thank you for your valuable observation/comment. In the previous research in our laboratory, it was found through orthogonal experiments that the best light exposure affecting the synthesis of functional metabolites of longan EC was blue light. In addition, blue light treatment could increase the proliferation rate of longan embryonic callus. However, red light treatment could not achieve optimal accumulation of functional metabolites in longan EC.BXs compounds are secondary metabolites. Therefore, the expression pattern of Bx under red light was not detected in this experiment. However, thank you very much for your valuable comments, and in the next experiments I will consider performing red light treatment of longan EC to explore the expression pattern of Bx. In addition, the hormone treatment and light treatment in this experiment are two separate experiments and are therefore not analyzed and discussed together.

Round 2
Reviewer 3 Report
Comments and Suggestions for Authors
Dear authors, I have review your manuscript in the second round, in the pdf manuscript yow will find some observations that are required to complete the revision.

Author Response
Thank you very much for your input, which has benefited me a lot. In response to your comments, I have made the corresponding changes in the manuscript and made the following explanations based on the questions you raised.
- According to the heatp map, red color means 1.0 to 1.5 Log2, Log10???, in anycase. means higher expression. What was the calibrator, the RQ of the calibrator is normally 1. If so, your account number of genes upregulated is totally different than you explained in manuscript.Please add a table of expression levels, what was the calibrator.
Thank you for your valuable observation/comment. In the heat map, red indicates higher expression levels and blue indicates lower expression levels. The expression level is presented in the form of log2 (FPKM+1) values. The heat map is based on the FPKM values in the transcriptome data under various treatments, visualized by using TBtools-II v2.080 software, and the parameters are set as follows: Log Scale=Log2, Row Scale, Scale Method=Normalized. The FPKM values of DlBx gene in the early somatic embryogenesis stage of longan and various treatments were provided in Suppl. Table S8. I have made changes in part 4.6 of the manuscript.
- What units did you use to know the expression? Relative, Log2, Log10?
Thank you for your valuable observation/comment. The expression level of each gene was calculated as FPKM, transformed as log2(FPKM+1).
- STRING database v12.0,and what confidence level you used? 0.4, 0.7., 0.9?
Thank you for your valuable observation/comment. The confidence level I used was 0.7. I have made changes in part 4.5 of the manuscript.

Round 3
Reviewer 3 Report
Comments and Suggestions for Authors
Dear authors, I am asking you to explain your data derived from FPKM into Log2 and the heat map elaboration. The data you may add is in the last page of the pdf manuscript.

Author Response
Due to our negligence and misunderstandings with the PLANTS journal editors, we did not upload the responds and revised manuscripts. I'm very sorry for that. This time, I answered the questions you asked and revised them in the manuscript. Thank you very much for your observation/comment on my manuscript.
- What concentration of 2,4-D was used? Kinetin concentration?
Thank you for your valuable observation/comment. The concentrations of 2,4-D and Kintin are 1 mg/L and 5 mg/L, respectively. I have made corresponding revisions in manuscript 2.5.
- Dear authors, I am asking you to explain your data derived from FPKM into Log2 and the heat map elaboration. The data you may add is in the last page of the pdf manuscript.
Thank you for your valuable observation/comment. The raw FPKM values used in this article are provided in Suppl. Table S8. I've added the data from FPKM to Log2 to the last page in the manuscript (Table S9). Here's how a heatmap is made (Take the expression heat map of DlBx under different hormone treatments as an example):
In the first step, prepare the expression matrix of DlBx under different hormone treatments, that is, the FPKM value of DlBx under different hormone treatments.
|
|
2,4-D |
2,4-D+KT |
KT |
MS |
|
DlBx1 |
67.16 |
69.50333333 |
83.9 |
85.18666667 |
|
DlBx3a |
0.16 |
0.3 |
0.033333333 |
0.08 |
|
DlBx3b |
0.126666667 |
0.126666667 |
0.016666667 |
0.03 |
|
DlBx3f |
0.123333333 |
0.106666667 |
0.156666667 |
0.22 |
|
DlBx3g |
99.33333333 |
94.1 |
48.41333333 |
39.81666667 |
|
DlBx4b |
0.046666667 |
0.08 |
0 |
0 |
|
DlBx6a |
0.18 |
0.403333333 |
0.173333333 |
0.32 |
|
DlBx6b |
12.42333333 |
13.29666667 |
10.69666667 |
10.3 |
|
DlBx6c |
5.233333333 |
4.21 |
3.51 |
2.706666667 |
|
DlBx6d |
0.07 |
0 |
0 |
0.083333333 |
|
DlBx6e |
0.863333333 |
1.326666667 |
1.31 |
0.97 |
|
DlBx6f |
0.4 |
0.903333333 |
0.306666667 |
0.28 |
|
DlBx6g |
2.033333333 |
1.626666667 |
1.613333333 |
1.406666667 |
|
DlBx6h |
4.04 |
3.223333333 |
4.016666667 |
2.563333333 |
|
DlBx7a |
0.21 |
0.253333333 |
0.133333333 |
0.193333333 |
|
DlBx7b |
0.21 |
0.253333333 |
0.133333333 |
0.193333333 |
|
DlBx7d |
0 |
0.16 |
0.316666667 |
0.253333333 |
|
DlBx7e |
0.036666667 |
0 |
0.143333333 |
0.133333333 |
|
DlBx8 |
3.633333333 |
4.106666667 |
3.736666667 |
3.556666667 |
|
DlBx9a |
0.053333333 |
0 |
0.016666667 |
0 |
|
DlBx9b |
3.633333333 |
4.106666667 |
3.736666667 |
3.556666667 |
The second step is to open the HeatMap tool of TBtools-II v2.080 (Graphics→HeatMap Illustator→HeatMap), and enter the expression matrix into the first box (Set input ID list), and click ‘Start’ to get an initial graph.

The third step is to click 'Show Control Dialog' above to adjust the advanced properties of the image, log2 is selected for logarithmic processing of the heatmap, and normalize and cluster by row. The data shown in the figure is the original FPKM value, with two decimal places selected (I have added this in the legend section of Figure 6). This is shown in the figure below:

select Log2, the displayed value is Log2 (FPKM+1)

→ select ‘Row Scale’ and ‘Cluster Rows’

→ select ‘Show Ori Value’

Round 4
Reviewer 3 Report
Comments and Suggestions for Authors
Dear authors, thank you very much for your answers, the manuscript is ready to be publish.